# The prognostic significance of additional localized treatment to primary lesion in patients undergoing hormone therapy for metastatic hormone-sensitive prostate cancer: A systematic review and meta-analysis

Yuta Yamada[1], Fumihiko Urabe[2]*, Shoji Kimura[2,3], Kosuke Iwatani[2,3], Naoki Kimura[1], Jun Miki[2,3], Takahiro Kimura[2], Haruki Kume[1]

1 Department of Urology, Graduate School of Medicine, The University of Tokyo, Tokyo, Japan,
2 Department of Urology, The Jikei University School of Medicine, Tokyo, Japan, 3 Department of Urology, The Jikei University Kashiwa Hospital, Chiba, Japan

* furabe0809@gmail.com

## Abstract

### Background

We aimed to compare the prognostic values of 'localized treatment to the primary lesion (LT) plus hormone therapy (HT)' versus 'HT alone' in metastatic hormone-sensitive prostate cancer (mHSPC).

### Methods

We conducted a systematic search through the databases of PubMed®, Web of Science®, and Cochrane library® in April 2023 based on the PRISMA (Preferred Reporting Items for Systemic Reviews and Meta-Analyses) statement. A pooled meta-analysis was performed to assess the prognostic differences between LT + HT and HT alone according to randomized and non-randomized controlled studies (RCTs and NRCTs, respectively).

### Results

The search identified three RCTs and eight NRCTs. In RCTs, LT did not show prognostic benefits regarding biochemical-failure free rate nor overall survival (OS), although in patients with low tumor burdens, the LT + HT group showed better OS (HR: 0.68, 95% CI: 0.54–0.86). In the NRCTs, the LT+HT group showed superior progression-free survival (hazard ratio (HR): 0.42, 95% confidence interval (CI): 0.21–0.87), cancer-specific survival (HR: 0.39, 95% CI: 0.20–0.76), and OS (HR: 0.63, 95% CI: 0.57–0.69) to the HT alone group. In addition, better OS was observed in the LT +HT group regardless of the type of treatment modality for LT; radical prostatectomy (HR: 0.52, 95% CI: 0.39–0.69), radiotherapy (HR: 0.63, 95% CI: 0.56–0.71) in NRCTs.

**Data Availability Statement:** All relevant data are within the manuscript and its Supporting Information files.

**Funding:** The author(s) received no specific funding for this work.

**Competing interests:** The authors have declared that no competing interests exist.

## Conclusions

LT to the primary lesion in metastatic hormone-sensitive prostate cancer may provide prognostic benefits and especially in patients with low tumor burden.

## Introduction

Androgen deprivation therapy (ADT) has been the first-line therapy for metastatic hormone-sensitive prostate cancer (mHSPC) for many years [1]. However, several randomized studies (RCTs) suggest that intensified treatment that may provide additional anti-cancer effect leading to better prognosis [2–4]. The CHAARTED trial proved superior castration-resistant cancer-free survival in patients treated with docetaxel and ADT [2]. The STAMPEDE [3] and LATITUDE [4] trials also showed better prognostic outcomes with combined treatment using abiraterone acetate/prednisone and ADT. Moreover, a recent RCT, ARASENS trial showed better prognostic outcomes when darolutamide and docetaxel were added to standard ADT [5]. On the other hand, localized treatment to the primary lesion or metastasis directed therapy in mHSPC has also been suggested [6,7]. The STAMPEDE arm H and HORRAD trials and meta-analysis using these two studies have shown OS benefits by performing radiation therapy (RT) to the prostate in mHSPC patients [8–10]. However, there is no meta-analysis evaluating the clinical impact of LT in mHSPC patients, although cytoreductive radical prostatectomy (RP) has also gained widespread use in part by the introduction of robot-assisted radical prostatectomy [11]. In this systematic review, we investigated the prognostic value of localized treatment in mHSPC patients.

## Materials/Subjects and methods

The protocol has been registered in the International Prospective Register of Systematic Reviews database (PROSPERO: CRD42023430905). The Preferred Reporting Items for Systematic Reviews and Meta-Analyses (PRISMA) checklist is reported in S1 Table.

### Literature search and inclusion and exclusion criteria

The systematic review and meta-analysis were carried out according to the PRISMA statement [12] and the Cochrane Handbook for Systematic Reviews of Interventions [13]. A literature search of electronic databases (MEDLINE, Web of Science, Cochrane Library) was performed on the April 10, 2023. The initial screening on the titles and abstracts was performed to identify eligible studies that was appropriate for the topic of this study. In addition, all full text papers were assessed and excluded with reasons when deemed inappropriate. Two reviewers carried out this process independently. Disagreements were resolved by a third party. The following string terms were used: *(((metastatic hormone sensitive prostate cancer) OR (metastatic castration sensitive prostate cancer)) AND ((((localized therapy) OR (radical prostatectomy)) OR (ablation therapy)) OR (radiation therapy))) AND ((androgen deprivation) OR (androgen receptor axis targeted agent)).*

The largest or most recently published study was included whenever there were multiple articles written by the same authors based on a similar patient cohort or clinical study. Review articles, letters, editorials, comments and meeting abstracts were excluded. References of included manuscripts were also investigated for additional studies of interest.

## Data extraction

Two authors (Y.Y. and S.K.) independently extracted the data. Data on clinical parameters such as age, prostate specific antigen (PSA), cT stage and prognosis were collected from the searched articles. The primary outcome of interest was overall survival (OS) and the secondary outcomes of interest were cancer-specific survival (CSS), progression-free survival (PFS), and metastasis-free survival (MFS). All discrepancies regarding data extraction were resolved by consensus or finally decided by Delphi consensus with other authors.

## Statistical analysis

Regarding, meta-analysis, we analyzed the data from the RCTs and the NRCTs separately to identify any potential bias arising from study design. We performed a formal meta-analysis of PFS, CSS and OS using hazard ratios (HRs) with their 95% confidence intervals (CIs) extracted from selected articles directly to calculate pooled HRs. Statistical heterogeneity among studies were calculated using the $I^2$ statistics. The Chi-square test and $I^2$ statistics with significances set at $p < 0.10$ and $I^2 < 50\%$, respectively, were used to assess statistical heterogeneity among the studies. If there was a lack of heterogeneity, fixed-effects models were used for meta-analysis. Random-effects models were used in cases of heterogeneity. To evaluate publication bias, Egger linear regression and funnel plots were examined. Statistical analyses were performed using Stata 15.0 statistical software (Stata Corp, College Station,TX).

## Risk of bias assessment

The quality and risk bias were assessed by the Cochrane 'Risk of bias tool for RCTs' for RCTs [13] (S1A Fig) and 'Risk of Bias In Non-Randomized Studies -of Interventions (ROBINS-I) for NRCTs (S2 Table) [14]. Two authors (Y.Y. and S.K.) independently assessed the risk of bias in each study. All discrepancies between the two assessments were resolved by a consensus between the two authors and the supervisor (F.U.).

# Results

## Study selection and characteristics

The initial search identified a total of 372 articles (Fig 1). Eight articles that were identified from reference lists of the original search were added. After removing duplicates, 380 articles were identified for further processing. Subsequently, 363 articles were excluded after title and abstract assessment, respectively. Finally, 11 studies that reported the prognosis in both treatment groups (LT+HT versus HT alone) were included for qualitative and quantitative analyses after full-text reading [8,9,15]. The general characteristics of the eligible studies are summarized in Table 1. This systematic review included three RCTs [8,9,15] comprising 2,693 patients and eight NRCTs [16–23] comprising 8137 patients published between 2010 and 2023.

## Meta-analysis

**Comparison of prognosis between LT+HT group and HT alone group.**   In the RCTs, LT showed no prognostic benefits regarding BFS (HR: 0.60, 95% CI: 0.35–1.05) nor OS (HR: 0.87, 95% CI: 0.63–1.19) (Fig 2A and 2B). In the NRCTs, the LT+HT group showed better PFS (HR: 0.42, 95% CI: 0.21–0.87), CSS (HR: 0.39, 95% CI: 0.20–0.76), and OS (HR: 0.63, 95% CI: 0.57–0.69) (Fig 3A–3C).

**Subgroup-analyses stratified by tumor burden in RCTs.**   In patients with low tumor burden, the LT + HT group showed better OS (HR: 0.68, 95% CI: 0.54–0.86) (Fig 4A). In patients with high tumor burden, LT + HT did not result in better OS (HR: 1.07, 95% CI: 0.92–1.24) (Fig 4B).

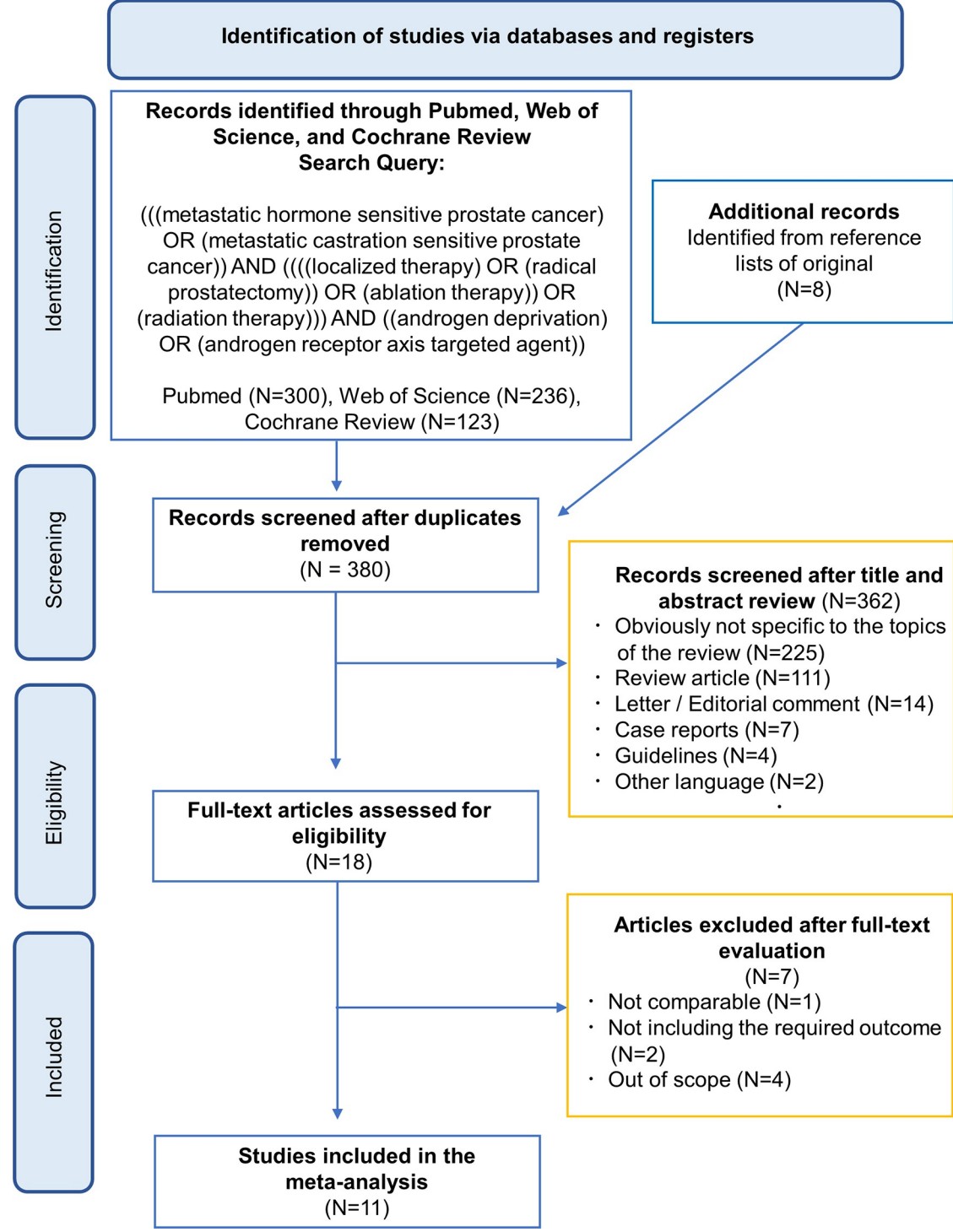

**Fig 1. PRISMA flow-chart of the systematic review and meta-analysis.**

## Subgroup-analyses stratified by types of LT in NRCTs

We also performed a subgroup-analyses depending on treatment modality. When RT was performed as LT, better OS was also observed (HR: 0.63, 95% CI: 0.56–0.71) (Fig 5). Better OS

**Table 1. Characteristics of studies included in the systematic review and meta-analysis.**

| Author | Year | Country | Study design | Procedure | No. pts. | Survival analysis | Ages, years median/mean | PSA ng/mL median/mean |
|---|---|---|---|---|---|---|---|---|
| Parker | 2018 | UK | RCT | RT + ADT | 1032 | OS, CSS, PFS, and MPFS | 68 (63–73) | 97 (33–313) |
| | | | | ADT | 1029 | | 68 (63–73) | 98 (30–316) |
| Boeve | 2019 | Netherlands | RCT | RT + ADT | 216 | OS, BFFS, | 67 (62–71) | 125 (48–433) |
| | | | | ADT | 216 | | 67 (61–71) | 149 (50–483) |
| Dai | 2022 | China | RCT | RP + ADT | 85 | OS, BFFS, RPFS | 67 (62–71) | 90 (35–236) |
| | | | | RT + ADT | 11 | | | |
| | | | | ADT | 100 | | 69 (64–73) | 102 (49–254) |
| Chi | 2021 | China | P | RP + ADT | 22 | OS, PFS, and RPFS | 69 (64.5–73) | 94.35(124.61) |
| | | | | ADT | 74 | | 70 (64.25–74) | 84.75(108.33) |
| Kim | 2019 | Korea | R | RP or RP + ADT | 219 | OS and CSS | 66.5 (61.0–71.8) | 69.2 (15.0–182.0) |
| | | | | ADT | 660 | | | |
| Morgan | 2021 | Canada | R | RT + ADT | 128 | OS | 75 (67–82) | 134.94 (39–500) |
| | | | | ADT | 282 | | 75 (70–83) | 56.6 (15.6–180.7) |
| Sheng | 2017 | China | R | Cryo + ADT | 23 | PFS | 68.1 ± 9.9 | 110.1 ± 35.02 |
| | | | | ADT | 26 | | 72.0 ± 4.7 | 98.42 ± 44.17 |
| Si | 2021 | China | R | RP + ADT | 27 | OS | 76.67 ± 9.66 | 28.93 (10.76–100) |
| | | | | ADT | 57 | | 76.42 ± 9.69 | 70.83 (26.08–100) |
| Jang | 2017 | Korea | R | RARP w/wo ADT | 38 | CSS and PFS | 65 (62–69) | 39.0 (15.0–84.5) |
| | | | | ADT | 41 | | 71 (67–76) | 50.0 (23.8–162.8) |
| Bhindi | 2017 | USA | PSM | RP + ADT | 79 | OS and CSS | 66 (SD 7) | 49.6 (17.4–86.0) |
| | | | | ADT | 79 | | 65 (SD 7) | 52.8 (30.7–103.0) |
| Rusthoven | 2016 | USA | R | RT + ADT | 538 | OS | 66 (59–74) | 66 (59–74) |
| | | | | ADT | 5844 | | 69 (61–78) | 69 (61–78) |

pts: patients, PSA: prostate specific antigen, pts: patients, UK: United Kingdom, RCT: randomized controlled trial, RT: radiation therapy, ADT: androgen deprivation therapy (includes surgical castration), OS: overall survival, CSS: cancer specific survival, PFS: progression free survival, MPFS: metastasis progression free survival, BFFS: biochemical failure free survival, USA: United States of America, R: retrospective study, Cryo: cryotherapy, RARP: robot-assisted radical prostatectomy, RP: radical prostatectomy, PSM: propensity score matched study.

(HR: 0.52, 95% CI: 0.39–0.69), CSS (HR: 0.28, 95% CI: 0.18–0.43), and PFS (HR: 0.57, 95% CI: 0.32–1.02) were also observed when RP was selected for LT (Fig 6A–6C).

## Discussion

The purpose of local therapy to the prostate in patients with advanced prostate cancer had been palliative use [24], until recently. Some reports have suggested that the local therapy may provide prognostic benefits [8,9]. It is widely known that there is interaction between the primary tumor and the distant metastasis, and that the primary tumor may secrete chemokines [25], growth factors [25], and extracellular vesicles [25–27] that may create the 'pre-metastatic niche' in distant metastatic sites or proliferation of metastatic cancer cells [28,29]. Removal of the primary tumor may avoid the tumor-promoting effect of the primary lesion and also prevent the forming of new metastases.

RP and RT may become a popular treatment option for local therapy in metastatic prostate cancer [8,9,15]. Jang et al. retrospectively reviewed the records of 79 patients with oligometastatic prostate cancer that were treated by either robot-assisted radical prostatectomy (RARP) or ADT [21]. Progression-free survival (PFS) and cancer-specific survival (CSS) were longer in RARP-treated patients (median PFS: 75 vs. 28 months, P = 0.008; median CSS: not reached vs.

## Randomized controlled studies

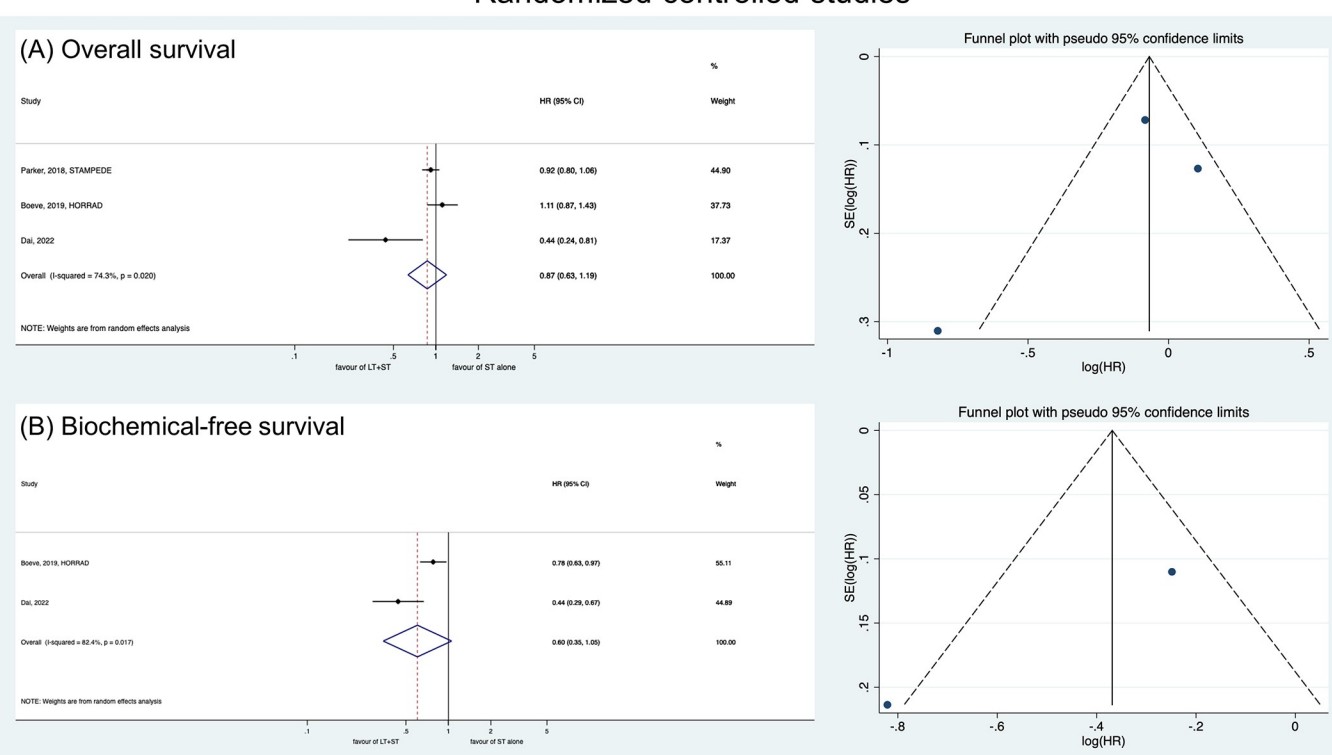

**Fig 2.** Forest plots showing the comparison of (A) overall survival and (B) biochemical-failure survival between LT+HT group and HT alone group in randomized controlled studies.

40 months, P = 0.002). On top of this, patients undergoing RARP showed fewer urinary complications than ADT-treated patients. A RCT by Dai et al. showed the clinical significance of local therapy in mHSPC [15]. Most of the patients in the local therapy group included RARP-treated patients (85 out of 96). In this study, patients treated with local therapy showed better OS (HR 0.44, 95%CI 0.24–0.81).

Recommended types of treatment may depend on high tumor volume (high risk) or high tumor burden which is defined as: ≥4 bone metastases including ≥1 outside vertebral column or pelvis and/or visceral metastasis in CHAARTED trial [2]; ≥4 bone metastases regardless of location or any visceral metastasis in STAMPEDE trial [3,8], and ≥2 high-risk features of: ≥3 bone metastases, presence of visceral metastasis, and ≥ISUP grade4 [ref.4]. In the CHAARTED trial, treatment intensification using ADT+ docetaxel improved OS in high volume disease (HR 0.60, 95%CI 0.45–0.81). On the other hand, in the STAMPEDE trial, RT to the prostate provided prognostic benefits in OS (HR 0.68, 95% CI 0.52–0.90), CSS (HR 0.65, 95% CI 0.47–0.90), and PFS (HR 0.80, 95% CI 0.63–1.01) in metastatic prostate cancer patients with low metastatic burdens [8]. Unfortunately, in patients with high metastatic burden, these prognostic benefits were not observed by the use of radiotherapy. Notably, older patients (≥70 years) received prognostic benefits from radiotherapy (HR 0.78, 95%CI 0.63–0.98). Taken together, intensification of systemic therapy maybe effective in patients associated with more widespread disease, while additional localized therapy to standard systemic therapy may provide survival benefit to a less-spread disease.

A recent RCT, the PEACE-1 study investigated the clinical significance of abiraterone with or without RT, in addition to standard of care (ADT alone or with docetaxel) in metastatic

## Non-randomized controlled studies

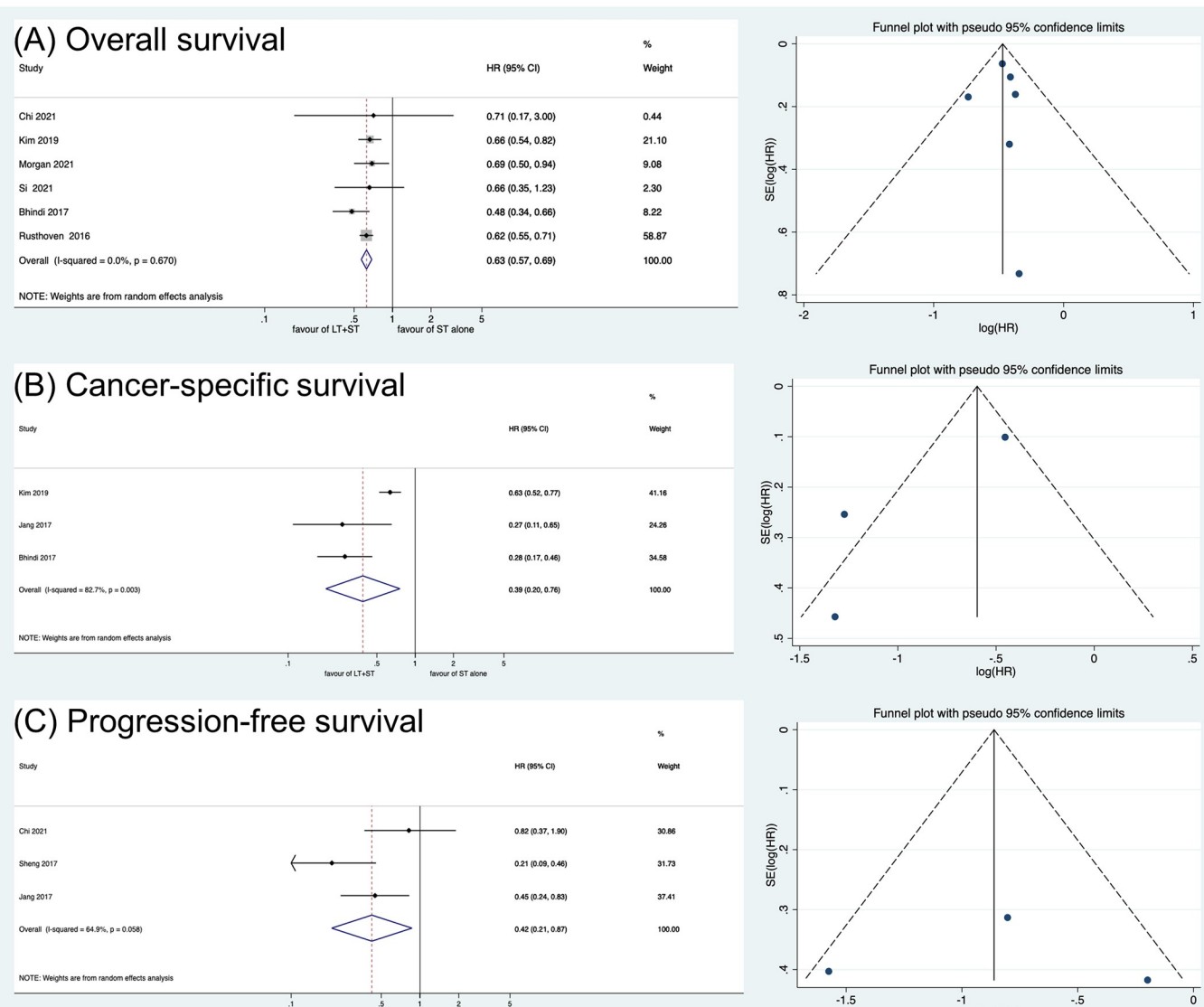

**Fig 3.** Forest plots showing the comparison of (A) overall survival, (B) cancer-specific survival, and (C) progression-free survival between LT+HT group and HT alone group in Non-randomized comparative studies.

castration-sensitive prostate cancer. This study was conducted with a 2 × 2 factorial design investigating the differences in prognosis among ADT alone or with docetaxel (standard of care; SOC), SOC plus radiotherapy (RT), SOC plus abiraterone, or SOC plus RT plus abiraterone [30]. Unfortunately, the analysis of this study was mainly focused on the use of abiraterone and thus the clinical significance of RT was not shown.

There are several limitations to point-out in this systematic review and meta-analysis. There were only 3 RCTs included with a relatively low number of participants. Therefore, we performed analysis separately in RCTs and NRCTS. The definition of 'tumor burden' is different since the term 'high tumor burden' in the CHAARTED study or the STAMPEDE study (arm H) is defined as 'four or more bone metastases with one or more outside the vertebral bodies or pelvis, or visceral metastases, or both' [2,3,8], while in the LATITUTE study, the

## Randomized controlled studies

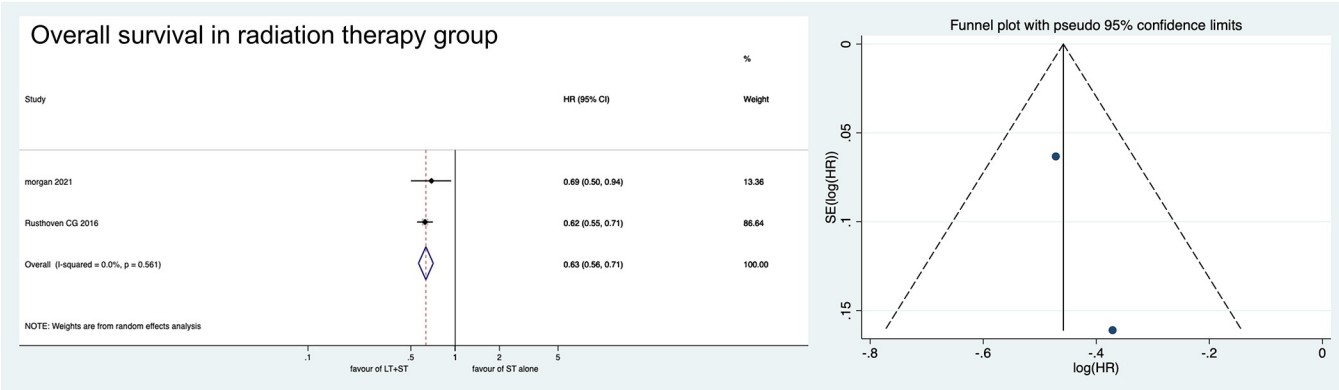

**Fig 4.** Forest plots showing the comparison of overall survival between LT+HT group and HT alone group in patients with (A) low tumor burden and (B) high tumor burden (randomized controlled studies).

term was 'high risk' which was defined as two or more high-risk features of the following: 1. three or more bone metastases, 2. presence of visceral metastasis, 3. ISUP grade 4 and over [4]. The sub-analysis regarding the tumor burden of the present study is not perfectly accurate in terms of the selected patients due to these varying definitions.

## Non-randomized comparative studies

**Fig 5. Forest plots showing the comparison of overall survival between LT+HT group and HT alone group in patients undergoing radiotherapy for local therapy.**

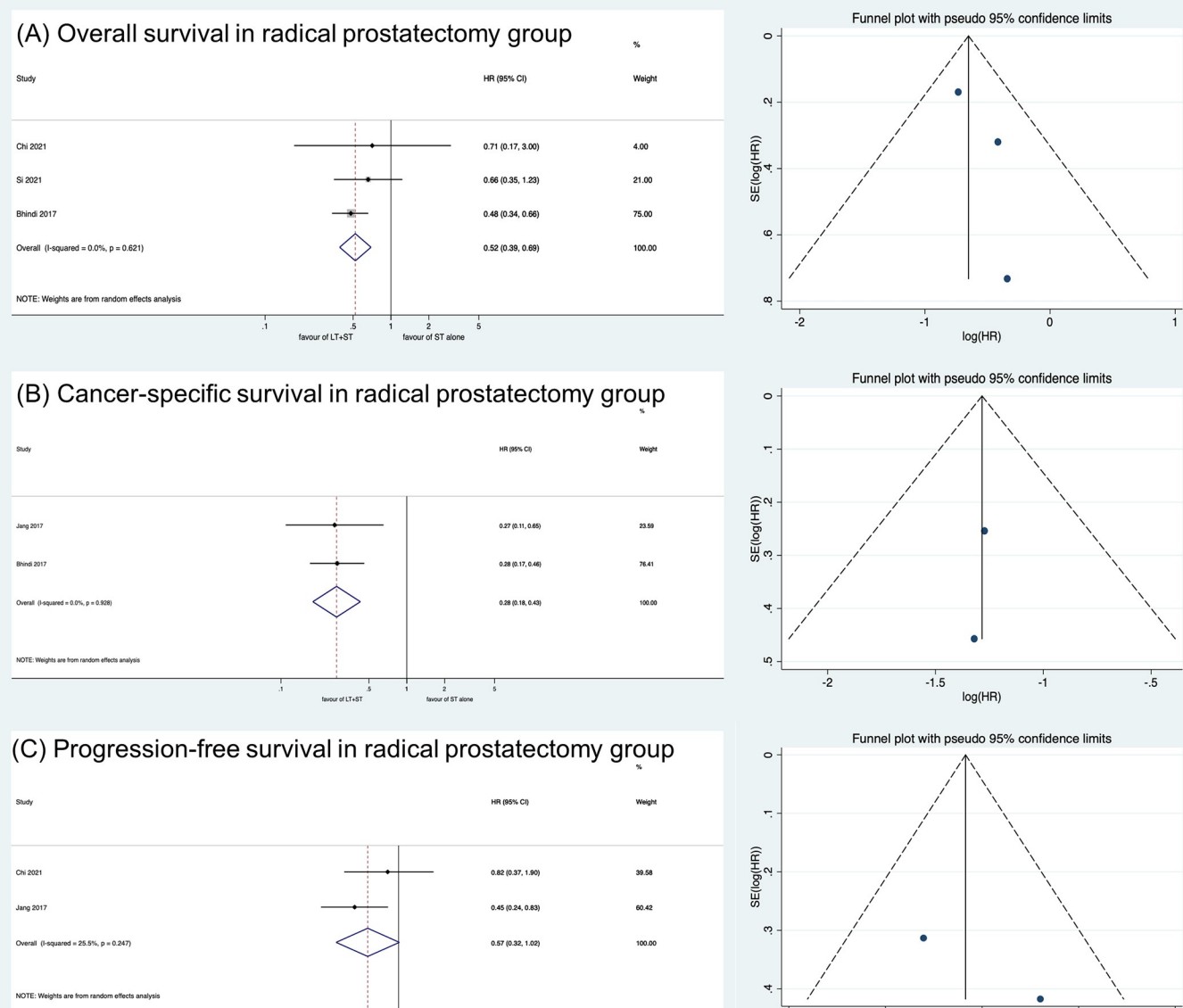

**Fig 6.** Forest plots showing the comparison of prognosis between LT+HT group and HT alone group in patients undergoing radical prostatectomy for local therapy (A) overall survival, (B) cancer-specific survival, (C) progression-free survival.

In conclusion, local therapy to the primary prostate cancer in combination with hormone therapy may provide prognostic benefits especially in patients with low tumor burden. Further studies are required to assess the clinical impact of RARP-treated patients in this clinical setting.

## Supporting information

**S1 Fig. Risk of bias assessment of the included RCTs.** RCTs: randomized controlled studies. (DOCX)

**S1 Table. PRISMA-2009-checklist.** PRISMA: Preferred Reporting Items for Systematic Reviews and Meta-Analayses.
(XLSX)

**S2 Table. Risk of bias assessment for NRCTs (ROBINS-I).** NRCTs: non-randomized comparative studies, ROBINS-I: Risk of Bias In Non-Randomized Studies of Intervention.
(XLSX)

## Author Contributions

**Conceptualization:** Yuta Yamada, Fumihiko Urabe, Shoji Kimura, Kosuke Iwatani, Naoki Kimura, Jun Miki, Takahiro Kimura, Haruki Kume.

**Data curation:** Yuta Yamada, Fumihiko Urabe, Shoji Kimura, Kosuke Iwatani, Naoki Kimura.

**Formal analysis:** Yuta Yamada, Fumihiko Urabe, Shoji Kimura, Kosuke Iwatani.

**Investigation:** Fumihiko Urabe.

**Methodology:** Yuta Yamada, Fumihiko Urabe.

**Supervision:** Fumihiko Urabe, Jun Miki.

**Writing – original draft:** Yuta Yamada, Fumihiko Urabe, Shoji Kimura, Kosuke Iwatani.

**Writing – review & editing:** Naoki Kimura, Jun Miki, Takahiro Kimura, Haruki Kume.

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
