## [Decision Letter · Decision Letter 0]

8 May 2024

PONE-D-23-44163The prognostic significance of additional localized treatment to primary lesion in patients undergoing hormone therapy for metastatic hormone-sensitive prostate cancer: a systematic review and meta-analysisPLOS ONE

Dear Dr. Urabe,

Thank you for submitting your manuscript to PLOS ONE. After careful consideration, we feel that it has merit but does not fully meet PLOS ONE’s publication criteria as it currently stands. Therefore, we invite you to submit a revised version of the manuscript that addresses the points raised during the review process.

**Reviewer 1.**Management of metastatic hormone/castration sensitive prostate cancer has evolved over the last decade. A series of randomized clinical trials have shown that the addition of systemic chemotherapy (docetaxel) and androgen-receptor signaling pathway inhibitors to androgen deprivation therapy (ADT) improvement in cancer related outcomes.

Definitive local therapy is often incorporated in the management. Relief of symptoms or prevention of symptoms, elimination of tumor promoting effects and development of new metastases are cited as justifications for definitive local therapy. However, robust data from randomized clinical trials evaluating the effect of local therapy on improvement in cancer related outcomes have not been reported.

Yamada and coauthors are to be commended for submitting this manuscript which seeks to address this unmet need. The authors report the specifics of the literature search, data extraction and selection of clinical of 11 clinical trials (three randomized control trials and non-randomized trials).

In the 3 randomized clinical trials, with the STAMPEDE trial reported by Parker et al in the Lancet being the largest with nearly 2000 subjects, in unselected subjects local therapy (radiation) did not reveal improvement in biochemical failure-free or overall survival. Pooled results of the non-randomized trials did reveal improvement in progression-free survival, cancer specific and overall survival. Subset analysis showed that the benefit/improvement in cancer related outcomes were noted only in subjects with low tumor burden. This is undoubtedly due the fact that the extent of tumor burden will influence cancer related outcomes. Nevertheless, the results will be helpful to clinicians in identifying patients who are likely to benefit from definitive local therapy. The paper meets an unmet need.

The data presented stand on their merit. However, since English is unlikely the primary language of the authors recommend professional copyediting.**Reviewer 2.**This study evaluated the prognostic impact of "localized treatment to the primary lesion (LT) plus hormone therapy (HT)" versus "HT alone" in metastatic hormone-sensitive prostate cancer. The analysis is well-conducted, presenting robust evidence supporting the combined treatment approach, particularly in patients with a low tumor burden. However, the conclusion section only mentions "LT," which does not align with the central findings that emphasize the benefits of combining LT with HT. The authors should revise the conclusion to clearly state that the combination of LT+HT contributes to improved prognoses.

We look forward to receiving your revised manuscript.

Kind regards,

Mohamad Abou Chakra, MD

Academic Editor

PLOS ONE

Journal Requirements:

2. Thank you for stating the following in your Competing Interests section: "None"

Reviewers' comments:

Reviewer's Responses to Questions

**Comments to the Author**

1. Is the manuscript technically sound, and do the data support the conclusions?

Reviewer #1: Yes

Reviewer #2: Yes

2. Has the statistical analysis been performed appropriately and rigorously? 

Reviewer #1: I Don't Know

Reviewer #2: Yes

3. Have the authors made all data underlying the findings in their manuscript fully available?

Reviewer #1: Yes

Reviewer #2: Yes

4. Is the manuscript presented in an intelligible fashion and written in standard English?

Reviewer #1: No

Reviewer #2: Yes

5. Review Comments to the Author

Reviewer #1: Management of metastatic hormone/castration sensitive prostate cancer has evolved over the last decade. A series of randomized clinical trials have shown that the addition of systemic chemotherapy (docetaxel) and androgen-receptor signaling pathway inhibitors to androgen deprivation therapy (ADT) improvement in cancer related outcomes.

Definitive local therapy is often incorporated in the management. Relief of symptoms or prevention of symptoms, elimination of tumor promoting effects and development of new metastases are cited as justifications for definitive local therapy. However, robust data from randomized clinical trials evaluating the effect of local therapy on improvement in cancer related outcomes have not been reported.

Yamada and coauthors are to be commended for submitting this manuscript which seeks to address this unmet need. The authors report the specifics of the literature search, data extraction and selection of clinical of 11 clinical trials (three randomized control trials and non-randomized trials).

In the 3 randomized clinical trials, with the STAMPEDE trial reported by Parker et al in the Lancet being the largest with nearly 2000 subjects, in unselected subjects local therapy (radiation) did not reveal improvement in biochemical failure-free or overall survival. Pooled results of the non-randomized trials did reveal improvement in progression-free survival, cancer specific and overall survival. Subset analysis showed that the benefit/improvement in cancer related outcomes were noted only in subjects with low tumor burden. This is undoubtedly due the fact that the extent of tumor burden will influence cancer related outcomes. Nevertheless, the results will be helpful to clinicians in identifying patients who are likely to benefit from definitive local therapy. The paper meets an unmet need.

The data presented stand on their merit. However, since English is unlikely the primary language of the authors recommend professional copyediting.

Reviewer #2: This study evaluated the prognostic impact of "localized treatment to the primary lesion (LT) plus hormone therapy (HT)" versus "HT alone" in metastatic hormone-sensitive prostate cancer. The analysis is well-conducted, presenting robust evidence supporting the combined treatment approach, particularly in patients with a low tumor burden. However, the conclusion section only mentions "LT," which does not align with the central findings that emphasize the benefits of combining LT with HT. The authors should revise the conclusion to clearly state that the combination of LT+HT contributes to improved prognoses.

6. PLOS authors have the option to publish the peer review history of their article (what does this mean?). If published, this will include your full peer review and any attached files.

Reviewer #1: No

Reviewer #2: No

---

## [Author Response · Author response to Decision Letter 0]

17 May 2024

Dear Editor,

The comments of the reviewers are in italic and our response to these comments are in bold characters. The corrected parts are yellow-high-lighted in the new main-manuscript and the matched number of page and lines correspond to the page and line numbers of the new manuscript. 

↓↓↓↓↓

Reviewer 1.

Management of metastatic hormone/castration sensitive prostate cancer has evolved over the last decade. A series of randomized clinical trials have shown that the addition of systemic chemotherapy (docetaxel) and androgen-receptor signaling pathway inhibitors to androgen deprivation therapy (ADT) improvement in cancer related outcomes.

Definitive local therapy is often incorporated in the management. Relief of symptoms or prevention of symptoms, elimination of tumor promoting effects and development of new metastases are cited as justifications for definitive local therapy. However, robust data from randomized clinical trials evaluating the effect of local therapy on improvement in cancer related outcomes have not been reported.

Yamada and coauthors are to be commended for submitting this manuscript which seeks to address this unmet need. The authors report the specifics of the literature search, data extraction and selection of clinical of 11 clinical trials (three randomized control trials and non-randomized trials).

In the 3 randomized clinical trials, with the STAMPEDE trial reported by Parker et al in the Lancet being the largest with nearly 2000 subjects, in unselected subjects local therapy (radiation) did not reveal improvement in biochemical failure-free or overall survival. Pooled results of the non-randomized trials did reveal improvement in progression-free survival, cancer specific and overall survival. Subset analysis showed that the benefit/improvement in cancer related outcomes were noted only in subjects with low tumor burden. This is undoubtedly due the fact that the extent of tumor burden will influence cancer related outcomes. Nevertheless, the results will be helpful to clinicians in identifying patients who are likely to benefit from definitive local therapy. The paper meets an unmet need.

The data presented stand on their merit. However, since English is unlikely the primary language of the authors recommend professional copyediting.

→Thank you for the comment and advice. We had our manuscript proofread by a native speaker. The English proofreading certificate is also uploaded.

Based on the advice of the English proofreading, we corrected parts of the manuscript as below.

“The” is added in the manuscript in multiple parts of the manuscript as mentioned below.

page1 line15; page2 line48, 50, 51; page3 line97; page4 line124; page5 line147, 148 

Arabic numerals were corrected to English alphabets in

Page2 line46, page3 line84, page4 line143, and page6 line203.

Singular nouns that should be plural were corrected in 

Page 2 line47: “burdens”

Page 3 line 84: “trials”

Page 3 line 113: “articles”

Page 5 line186: “burdens”

Inappropriate expression of abbreviation was also corrected in

Page 2 line 82: “ADT”

Page 4 lines 120-121: “hazard ratios (HRs)”

Page 4 line130 and 131: “Supplementary”

Page 5 line 159: “RP”

Page 5 line 171: “RP and RT”

Page 5 line 172: “prostate cancer”

Page 5 line 173: “ADT”

Page 5 line 183: “docetaxel”

Page 6 line 184: “RT”

Page 6 line185: “CSS” and “PFS”

Page 6 line 192: “RT”

Other gramattical errors were checked and corrected to

Page 1 line 25: “short title”

Page 2 lines 39-40: “of ‘localized treatment to the primary lesion (LT) plus hormone therapy (HT)’ versus ‘HT alone’ in metastatic hormone-sensitive prostate cancer (mHSPC).”

Page 3 line77: “suggest that”

Page 3 line 82: “were” 

Page 3 line 98 “of”, line99: “April 10”

Page 3 line 106: “whenever”

Page 4 lines 119-120: “identify any”

Page 4 line 123 “significances”

Page 4 line 131: “assessed”

Page 4 line 132: “between the two assessments”

Page 4 line 139: “processing. Subsequently”

Page 5 line 147: “showed no”

Page 5 line159: “were” 

Page 5 line 165: “until recently. Some reports have”

Page 5 line 170: “the forming of new metastases”

Page 6 line 192: “investigated”

Page 6 lines 205-206: “due to these varying definitions”

Reviewer 2.

This study evaluated the prognostic impact of "localized treatment to the primary lesion (LT) plus hormone therapy (HT)" versus "HT alone" in metastatic hormone-sensitive prostate cancer. The analysis is well-conducted, presenting robust evidence supporting the combined treatment approach, particularly in patients with a low tumor burden. However, the conclusion section only mentions "LT," which does not align with the central findings that emphasize the benefits of combining LT with HT. The authors should revise the conclusion to clearly state that the combination of LT+HT contributes to improved prognoses.

→Thank you for pointing out. We corrected the sentence to “In conclusion, local therapy to the primary prostate cancer in combination with hormone therapy may provide prognostic benefits especially in patients with low tumor burden.” in page6 lines 207-208.

---

## [Editor Report · Decision Letter 1]

22 May 2024

The prognostic significance of additional localized treatment to primary lesion in patients undergoing hormone therapy for metastatic hormone-sensitive prostate cancer: a systematic review and meta-analysis

PONE-D-23-44163R1

Dear Dr. Fumihiko Urabe,

We’re pleased to inform you that your manuscript has been judged scientifically suitable for publication and will be formally accepted for publication once it meets all outstanding technical requirements.

Kind regards,

Mohamad Abou Chakra

Academic Editor

PLOS ONE
---

## [Editor Report · Acceptance letter]

30 May 2024

PONE-D-23-44163R1 

PLOS ONE

Dear Dr. Urabe, 

I'm pleased to inform you that your manuscript has been deemed suitable for publication in PLOS ONE. Congratulations! Your manuscript is now being handed over to our production team.

Kind regards, 

on behalf of

Dr. Mohamad Abou Chakra 

Academic Editor

PLOS ONE